# Analyzing Trajectories of Acute Cigarette Reduction Post-Introduction of an E-Cigarette Using Ecological Momentary Assessment Data

**DOI:** 10.3390/ijerph19127452

**Published:** 2022-06-17

**Authors:** Alexandra Guttentag, Tuo-Yen Tseng, Donna Shelley, Thomas Kirchner

**Affiliations:** 1Department of Epidemiology, New York University School of Global Public Health, New York, NY 10003, USA; tom.kirchner@nyu.edu; 2Department of Health, Behavior and Society, Johns Hopkins Bloomberg School of Public Health, Baltimore, MD 21205, USA; tuoyen@jhu.edu; 3Department of Public Health Policy and Management, New York University School of Global Public Health, New York, NY 10003, USA; donna.shelley@nyu.edu; 4Departments of Epidemiology and Social and Behavioral Sciences, New York University School of Global Public Health, New York, NY 10003, USA

**Keywords:** electronic cigarettes, longitudinal analysis, ecological momentary assessment, Nagin clustering

## Abstract

Electronic cigarettes (ECs) may hold great potential for helping smokers transition off combustible cigarettes (CCs); however, little is known about the patterns that smokers follow when using an EC as a CC-substitute in order to ultimately reduce and quit smoking. Our primary aim in this study was to evaluate whether common patterns of CC use exist amongst individuals asked to substitute an EC for at least half of the CCs they would normally smoke. These patterns may elucidate the immediate switching and reduction behaviors of individuals using ECs as a reduction/cessation tool. This analysis uses data from a randomized controlled trial of 84 adult smokers assigned to receive either 4.5% nicotine or placebo (0% nicotine) EC. Participants were advised to use the EC to help them reach a 50% reduction in cigarettes-per-day (CPD) within 3 weeks. Longitudinal trajectory analysis was used to identify CPD reduction classes amongst the sample; participants clustered into four distinct, linear trajectories based on daily CC use during the 3-week intervention. Higher readiness to quit smoking, prior successful quit attempts, and lower baseline CC consumption were associated with assignment into “more successful” CC reduction classes. ECs may be a useful mechanism to promote CC reduction. This study demonstrates that a fine-grained trajectory approach can be applied to examine switching patterns in the critical first weeks of an attempt.

## 1. Introduction

Combustible cigarette (CC) smoking is the leading cause of preventable death worldwide [1]. Despite decades of educational campaigns on the harms of smoking and the promotion of smoking cessation programs and aids, an estimated 20% of the world and 14% of the U.S. population are current CC smokers [2]. Approximately half of these smokers try to quit annually [3]; however, 80% of smokers attempting to quit relapse within the first month [4]. Electronic cigarettes (ECs) are battery-powered nicotine delivery systems that researchers and harm-reduction advocates are currently exploring as a way to help smokers reduce and quit smoking CCs. While the scientific literature has suggested that, in certain cases, ECs can be a helpful support system for adult smokers transitioning away from CCs, many of the trials evaluating ECs as a CC cessation tool are point-prevalent in nature [5,6,7,8,9] and thus cannot evaluate the nuances that arise during a naturalistic switching process. Although cross-sectional studies have shown EC users to be more likely to achieve abstinence or smoking reduction versus other forms of nicotine replacement therapy (NRT) [9,10,11], there is limited literature on the actual reduction processes that occur as smokers attempt to reach complete CC abstinence while utilizing ECs. In particular, there is a gap in the literature investigating the behavioral changes that occur in the acute timeframe after a smoker begins displacing CCs or otherwise using ECs instead of CCs.

The objective of this study was to evaluate the immediate CC reduction behavior of individuals using ECs to help them reduce smoking, with a particular interest in examining whether certain baseline characteristics (e.g., demographics, smoking behavior) in addition to drug assignment (nicotine versus placebo) affected class assignment. A priori, we hypothesized that smokers would fall into different reduction trajectory groups; some groups would have more quickly-reducing trajectories than others, and potentially some trajectory groups might follow a cubic pattern—cigarette smoking might reduce initially and then increase again towards the end of the study. Overall, the groups would share common characteristics related to baseline smoking behavior and certain demographics including age, number of years smoked, and cigarettes smoked per day at baseline.

We explored the switching trajectories that smokers in this reduction-challenge study followed and evaluated common characteristics within each trajectory class to understand how certain individuals may be more likely to experience one form of smoking behavioral change versus another. 

## 2. Materials and Methods

The data for this study came from *Assessing the use of electronic cigarettes (e-cigarettes) as a harm reduction strategy*, an IRB-approved, double-blind randomized controlled trial at the NYU Langone School of Medicine in New York City in 2014 [12]. Eligibility criteria has been described previously [12]; briefly, this study recruited smokers in early adulthood (ages 21–35) who were interested in using an EC to help them reduce their smoking by 50% within three weeks. Participants were all smokers using at least 10 CPD upon enrollment; participants were excluded if they had used an EC in the 14 days prior to enrollment or used any other tobacco products besides CCs in the last 30 days. Participants were randomized into nicotine and placebo EC devices. Ecological momentary assessment (EMA) data using short message service (SMS, or text messaging) were a key component of the study protocol. EMA involves repeated sampling to capture real-time data from participants [13,14]. For 21 days, participants received text messages four times daily at 4 h intervals, tailored to the participants’ schedules of waking and sleeping. The text messages asked about CC and EC consumption—specifically, how many CCs the participant had consumed since the last visit, how many separate times they used the EC, their craving level for CCs, and their satisfaction with the EC. 

Following study enrollment, each participant had 3 study visits, at weeks 1, 2, and 3, in addition to a 20–30 min call on smoking cessation tips with a tobacco cessation counselor. At study visits 1 and 3, participants provided all of their used ECs. Given the number of ECs provided, research staff calculated average ECs used per day, by dividing total ECs used by the number of days since the ECs were picked up, or, for week 3, since their week 1 visit. These numbers reflect the variables electronic cigarettes per day (ECPD) at week 1 and ECPD at week 3. Treatment was also included as a covariate—whether the participant was randomized to receive the 4.5% nicotine NJOY King Bold ECs or the placebo (0% nicotine) NJOY King Bold ECs. The NJOY King Bold was chosen for the study because of how closely the look and feel of the e-cigarette matched traditional, combustible cigarettes [12,15].

The following demographic variables included in the analyses were gathered from the baseline survey completed by participants: age, education, gender, and race/ethnicity.

Participants self-reported average number of CPD at baseline; this value, in addition to time of first cigarette of the day, was used to calculate the Heaviness of Smoking Index (HSI) for each participant at baseline (a score of 0–5, where 5 is most heavy) [16]. The Glover–Nilsson smoking behavioral dependence questionnaire was asked of each participant; individuals were classified as having a mild, moderate, or strong to very strong dependence [17]. Research staff ascertained baseline confidence to quit smoking by asking each participant, “How confident are you that you could quit smoking completely and stay quit (0 = not confident at all, 10 = very, very confident)?” Readiness to quit smoking was based on the readiness to quit ladder (scores of 1–8 apply to current smokers) [18]. Participants also reported whether they made at least one serious CC-cessation attempt in the prior year. 

We also included measures of smoking behavior in our analyses: participants self-reported their average CPD at baseline, week 1 and week 3. Baseline Heaviness of Smoking Index (HSI) was determined for each participant along with smoking behavioral dependence and readiness to quit smoking levels. 

The behavioral indicator used to develop the trajectory groups was CPD, gathered via EMA surveys over the course of the 21-day intervention. The initial text message in each EMA message sequence prompted, “since your last report, did you smoke an e-cigarette or cigarette?” If the participant answered that they had used a cigarette, they were then prompted to answer the following question: “how many cigarettes did you smoke?” Given this question, overall data on CPD are still collected even when participants miss a text prompt entirely. We conducted sensitivity analyses to evaluate whether the missing text messages impacted final results and we found that the missing text message prompts did not change our reported findings. 

A total of 84 study participants picked up ECs from study staff and began the text messaging part of the protocol; 79 participants completed the week 3 assessment. In order to maximize analytical sample size and avoid introducing biases from loss to follow up, we used an intent-to-treat approach and recoded all missing reduction outcomes as zeros; all participants who were missing the final study visit were indicated in the data as not having met the study reduction target of 50% reduction in CPD. Our final analytical sample size was *N* = 84. At each of the study visits, research staff also asked participants how many CCs they were smoking per day, on average. Based on reported number of cigarettes per day (CPD) at week 3, we determined whether each participant achieved a 50% smoking reduction from their baseline CPD levels. Individuals enrolled in the study were compensated for their participation; for study participation, participants received up to USD190. 

The sample was comprised of men (68%) and women (32%) of varying educational attainment (25% with a high school degree or less, 39% with some college, and 35% with college or post-graduate degrees). The mean age among participants at enrollment was 28.4. Non-Hispanic African American/Black participants made up 31% of the sample followed by Non-Hispanic White (30%), Hispanic of any race (25%), and other Non-Hispanic (14%). The mean number of cigarettes per day was 14.3 (SD 4.9). A total of 47% of the sample reported having made a serious quit attempt in the last year. More details on the study sample can be found in Tseng et al [12]. 

### Modeling Procedure

To carry out the trajectory analysis, we used Nagin Clustering [19,20,21]. Nagin clustering uses finite growth mixture models for outcomes measured longitudinally; essentially, this is a latent class model with a longitudinal outcome variable. It follows that trajectories may be grouped into “latent classes” of individuals that follow approximately the same temporal trajectory.

Using longitudinal, latent class modeling, such as Nagin Clustering, it is possible to categorize individuals into common classes of smoking behavior. When examining the way smoking behaviors evolve over time, it is useful to model smoking behavior classes as within-person trajectories. These trajectories, which reflect how an individual’s smoking behavior changes (or does not change) post-EC introduction, can help researchers understand the switching process and estimate the amount of CC displacement attributable to EC use. 

Given an individual’s data, in this case CPD over 21 days, Nagin Clustering determines the shapes of the (fixed number of) classes and the probability that a participant’s data were generated by each of the different trajectory groups available. These probabilities follow a multinomial logit function. The model is fit using maximum likelihood estimation, and the posterior probabilities of class membership may be used to assign each person to their most likely trajectory group. The variable used to construct the trajectories was summed daily CC use, based on the text messages sent four times daily asking about CC use. 

The modeling steps for this analysis generally followed the methods used in recent drug dependency (alcohol/nicotine) studies [22,23,24,25]. Firstly, we determined the number and shape of trajectory groups. Using the “traj” package in STATA [19], we assessed different trajectory models; number of groups tested ranged from 2 to 4. As the sample is small, this helped prevent a highly unequal distribution of participants into each of the groups. We assessed constant, linear, and quadratic polynomial orders. The best-fitting polynomial for each number of trajectory groups was determined by evaluating Bayesian Information Criterion (BIC), optimizing for lowest absolute value, while simultaneously ensuring that the coefficients of the lines for each of the trajectory groups were significant (at the *p* < 0.05 level), and evaluating proportion of individuals in each group (verifying that no one group had fewer than 10% of the total sample). We evaluated model fit using CPD as a raw (unmodified, continuous) variable and with a transformation (square root transformation) CPD; the trajectory fit was improved with a square root-transformed CPD variable. 

Using a multinomial logistic regression model with trajectory class assignment as the dependent variable, we evaluated the impact of individual-level characteristics on group assignment. Covariates were included in the final multinomial regression model based upon prior literature and evidence of intra-class differences in univariate multinomial regressions. Final model selection was guided by post-hoc likelihood ratio testing and R^2^. Specifically, we optimized for the most parsimonious model (wherein likelihood ratio testing of nested models was non-significant at the *p* < 0.05 level) and improvements in R^2^. 

## 3. Results

### 3.1. Determination of Optimal Number and Order of Trajectory Classes

We determined the optimal number of trajectories to be four, all with linear classes (slopes = −0.07, −0.06, −0.03, 0.004) (Figure 1). A total of 27% of the sample was classified as a group of “rapid reducers” (largest negative slope), 21% of the sample was classified as “moderate reducers”, 32% into group 3 “slow reducers”, and 19% into “maintainers”, with a slightly positive slope. The mean posterior probabilities for each class being assigned to its respective class ranged from 0.90 (rapid reducers) to 0.98 (moderate reducers), indicating strong classification [26,27]. Table 1 displays the equation of the lines for each trajectory group.

### 3.2. CPD Reduction Trajectories Post-EC Introduction

Table 2 displays the bivariate associations between trajectory class membership and individual-level variables. In subsequent analyses, the maintainer group served as the reference group. Multinomial logit models were used to evaluate the associations between selected variables and each trajectory group. We computed McFadden’s R^2^ and likelihood ratio tests to help identify the composite model of best fit.

### 3.3. Differences between Trajectory Classes

The target outcome of the study, a 50% reduction in CPD, was different across the trajectory groups (*p* = 0.001). Among the participants assigned to the maintainer group, 4% achieved a 50% smoking reduction at week 3 as compared to the entire sample; in the rapid, moderate, and slow reducers group the percentage reaching 50% CPD reduction ranged from 30–34%. When evaluating reduction success on the class-specific level, the following percentage of people in each class reached the 50% reduction threshold: 61% of the rapid reducers, 89% of the moderate reducers, 56% of the slow reducers, and 13% in the maintainer group (*N* = 2 out of 16 total). 

There were also statistically significant group-level differences in the baseline HSI (*p* < 0.05) and the targeted outcome of 50% CPD reduction at week 3 (*p* = 0.02). Those assigned to the rapid and moderate reducer classes had more negative slopes (implying more rapid reduction over time) and smaller starting values of CPD. These participants also had lower scores on the HSI than the overall sample mean HSI (2.6) and lower scores than those in the slow reducers or maintainer class. Across the four classes, the rapid reducers had the lowest mean HSI score (2.2) and the maintainers had the highest (2.9). 

### 3.4. Combustible Cigarette Cravings by Trajectory Class

Appendix A display changes in average EC use and average CC craving (average, aggregated on the daily level). At the start of the study, the maintainers reported the highest CC cravings of all four classes and maintained this same level throughout the study (craving level of approximately 5 out of 10). Rapid reducers reported similar craving levels at the start of the study (~4.2) and also maintained this level, with craving levels increasing very slightly over the study period, ending at an average craving level of 4.4 by day 21. Though slow reducers and maintainers reported similar CC craving levels at the start of the study, by day 21 the slow reducers had minimized levels to an average of 3.7. The moderate reducers reported the lowest CC craving levels at the start of the study out of all the classes and by day 21, the average daily CC craving level in the class was reduced by one-third—still the lowest of all four classes. 

### 3.5. Electronic Cigarette Use by Trajectory Class

For EC use, measured as the separate number of times that the EC was used each day, the rapid, moderate, and slow reducers had similar levels of EC use initially. Over the 21-day study period, all four classes decreased their daily EC use by varying degrees. Rapid reducers decreased EC use by over half, while moderate reducers decreased EC use by about 20%. Slow reducers also decreased over the course of the study, ending just below five times per day at week 3. The maintainers used the EC the least at the start of the study and decreased slightly over time, ending at around five separate EC use events per day on day 21. 

### 3.6. Class Assignment as Dependent Variable in Multinomial Logistic Regression

The following covariates were included in the final, composite logistic regression model (Figure 2): readiness to quit smoking score, HSI score, and drug assignment (nicotine versus placebo). The maintainer class served as the reference group. Controlling for readiness to quit smoking score and treatment assignment, participants with higher HSI scores were less likely to be in the rapid (RRR (relative risk ratio) = 0.74, 95% CI = 0.39–1.38) and moderate classes (RRR= 0.64, 95% CI = 0.34–1.16) versus the maintainers, and equally likely to be classified as a slow reducer (RRR = 1.01, 95% CI = 0.96–2.54). In this model, the heaviness of smoking appears to be negatively associated with more successful reduction groups, though the confidence intervals all included 1.0. participants who were “more ready” to quit smoking (based on readiness to quit smoking score measured at baseline) were more likely to be in the rapid reducers, moderate reducers, and slow reducer group as compared to the maintainer group. In this final model, randomization to nicotine was associated with a lower likelihood of classification into the rapid and moderate reduction groups versus the maintainer group and slightly increased RRR for classification into the slow reducer versus maintainer group; confidence intervals all included 1.0. 

## 4. Discussion

These analyses were exploratory in nature, yet our findings confirm that clustering may be an appropriate tool for analyzing EC- and CC-use behaviors. An EMA-based approach to understanding CC- and EC-usage behaviors can help elucidate distinct reduction patterns using a latent class analysis approach with time-varying factors. In these analyses, we found four distinct trajectory classes among the participants in the RCT. The classes were discriminated using CPD over 21 study days, and all classes followed a linear pattern. While three of the classes decreased over time with negative slopes (rapid reducers with a slope of β = −0.07, moderate reducers had a slope of β = −0.06, and slow reducers had a slope of β = −0.03), there was one class with a slightly positive increasing slope over time. The identification of this class, the maintainers, suggests that the EC device may not have provided a meaningful mechanism for CC replacement for these individuals, and may have had the opposite effect of increasing their CC consumption (slope of β = 0.004). 

The “maintainer” group had the largest intercept or starting CPD value of the four classes. Of the three classes with varying levels of “success” (based on negative linear slopes), those assigned to the rapid reducers class experienced the largest decrease in CPD over time, though the second-lowest starting CPD values. Heavier CPD at baseline (more than 10 CPD) was not statistically different across the four classes and was not a statistically significant predictor of class membership in the multinomial logit model. Thus, we cannot definitively conclude that smoking more CPD at baseline results in greater challenges to using ECs as a direct substitute for CCs; in fact, individuals using large quantities of CPD may also be using more ECs than their counterparts and potentially have greater EC uptake than others. These analyses also support the notion that there is not one uniform pattern for switching from CCs to ECs. Further research should evaluate whether heavier smoking is associated with less likelihood of reduction success through EC usage. 

Certain baseline characteristics may play important roles in participants’ trajectory assignment. Our models revealed that higher levels on the baseline readiness to quit scale were associated with more likelihood of placement in a “higher” (or more successful with CC reduction) trajectory. This is consistent with the literature that those with higher readiness to quit levels have strong self-efficacy related to quit ability and have stronger motivations to quit versus those with lower levels [28]. Other literature confirms that smokers who report higher levels of readiness to quit tend to understand the negative impacts of smoking and have, in general, negative views of smoking [29,30,31]. As supported by the literature, we found that having experienced a successful quit attempt in the prior 12 months was a predictor of a “more successful” reduction class [32].

It is not apparent from our analysis that demographic characteristics (including age, gender, educational attainment, and race/ethnicity) had an impact on trajectory class assignment. This may reflect the limited ages sampled in our population (restricted to ages 21–35; mean age was 28). At the time of this study, EC devices were still new and there were few social influences or marketing campaigns around the devices. In general, older age is associated with the likelihood of smoking cessation [33].

Nearly 60% of the analytical sample reached the 50% smoking reduction threshold at week 3; the percentage of individuals in each trajectory class who reached the 50% outcome ranged from 13% (the maintainer group) to 89% (moderate reducers), with 61% success among the rapid reducers and 56% among slow reducers. Individuals in both the treatment and placebo groups reached the 50% reduction threshold and there were no group-level differences across treatment assignments (*p* = 0.96). When controlling for potential confounders in the final multinomial regression model, those randomized to nicotine ECs only had higher RRR of classification into moderate reducers versus maintainer class, and those randomized to nicotine had lower RRRs for classification into rapid/moderate classes versus the maintainer class. However, there is evidence of a nicotine effect early in the study; for every day 1 through 7, participants assigned to nicotine had an average CPD that was lower than those assigned to the placebo group (on day 7, CPD was 1.1 for both placebo and nicotine groups). We observed a similar pattern at week 2 (for all classes except the rapid reducer class assigned nicotine who had lower average CPD versus those assigned placeboes) but not at week 3. This suggests that the impact of nicotine may be most salient in the first week following the initiation of reduction. While the similarities of an EC device to a CC may make the switch easier, the nicotine likely plays an additional role over just the sensorimotor act of vaping in the immediate switching timeline. Given the younger age range of our adult population (ages 21–35), there may be some appeal to using ECs as a novel or exciting product that is not a common sentiment with an older adult population—this may have resulted in this population using the ECs more regardless of drug assignment. 

The majority of studies evaluating whether ECs can be used as a means for smoking cessation focus on dual-endpoints [5,6,7,8,9]. This research is novel in that the data were collected daily for 21 days as a means to assess immediate smoking behavior outcomes as smokers use the EC to assist them with smoking reduction and cessation. The data collected were longitudinal, and aggregated from participant-completed SMS surveys. The surveys used a “coverage” strategy in which participants were asked to report on total CC/EC since the last time they reported, thus returning an enhanced “daily diary” capture of all behavior [14]. The findings from this study elucidate the patterns that smokers may generally follow as they rely on the EC to transition off of CCs. 

Findings from these analyses add important contributions to the study of harm reduction in smoking cessation research. Our analyses explored whether nicotine-based ECs may help participants reduce their CC use over placebo (0% nicotine) ECs, and we found that certain individual-level characteristics may impact acute success with an EC. Nevertheless, there is not a uniform experience in CC–EC switching that can be broadly applied to all populations; while there are similarities across groups, different individuals have unique reduction/switching patterns. 

This study has important limitations to consider. Given the small size of the sample, the study is underpowered to detect meaningful effect sizes between and within the four trajectory groups. Secondly, the device used in the RCT would be considered now a “first generation”, or early and less technologically advanced, EC device. It may not deliver as much nicotine or deliver the nicotine as efficiently as newer models, thus underestimating the potential impact of ECs on CC cessation. Another limitation relates to the RCT study design: as part of the study, participants were encouraged to use the EC to help them achieve this 50% smoking reduction and went through a smoking cessation counseling call. Thus, the findings may not be generalizable to the broader smoking population who are not actively interested in quitting and/or have not received explicit directions to use an EC to reduce their smoking by half. This also impacts our trajectory classifications, as some participants had already started reducing their baseline CPD consumption by the first day of the intervention period. The data used in these analyses came exclusively from self-reporting (with the exception of used ECs that were gathered at weeks 1 and 3 for cross-validation of the ECPD endpoints at weeks 1 and 3); thus, as is the case in all survey-based studies, there may be self-report biases. Lastly, while intent to treat was widely used in the literature, there are some limitations to this approach, such as the increase in the likelihood of Type II error [34,35]. 

The EMA approach and analyses also present some limitations. As part of the EMA protocol, participants were required to report, four times daily their CC and EC consumption and cravings and satisfaction levels; as aforementioned, there is the possibility of self-report biases here. However, EMA is thought to help reduce the impacts of self-report biases as it puts a lesser burden on participants to remember health behavior (as participants are asked multiple times daily instead of on a weekly basis, for example) [14,36,37]. One potential issue that arises with EMA data is missingness; participants were instructed to answer all text messages (further incentivized with monetary bonuses), but some text messages were still inevitably missed—applied EMA methodology typically accounts for non-100% compliance [38,39,40,41,42]. Based on our sensitivity analyses of the data, missing text messages did not impact our overall findings. Further, the format of the SMS survey followed a “coverage” approach or an enhanced version of “daily diary” data gathered in some EMA-based research [14]. With this approach, complete experiences and behavioral values are captured on a daily basis. In contrast to traditional EMA approaches in which SMS surveys may arrive at random times throughout the day and ask about instantaneous experience or behavior, the coverage approach allows for a complete aggregation of dual-use and dual-parallel behaviors [14]. Given the format of the SMS survey in this study, overall data on CPD are still collected even when participants miss a text prompt entirely.

Despite its limitations, the study has important strengths. Firstly, despite the limited size, the sample was diverse, with the Black and Latino sample population reaching over 50% of the total sample size. These populations face unique discrepancies in smoking cessation research as they are less likely to use any cessation aid and have more difficulties quitting [43,44,45,46,47] and Black and Latina women in particular face a higher burden of smoking-related morbidities [48]. Lastly, the methodology is novel: using clustering to study trajectories of CC to EC use in the acute switching phase has not been extensively studied. 

## 5. Conclusions

The trajectories described in these analyses reveal four distinct patterns that may take place when someone who is not ready to quit is given the tools and encouraged to begin using ECs to replace CCs, in a harm-reduction fashion. With dual-endpoint studies, researchers cannot view nuances in quit behavior and how patterns develop over time. Understanding features that impact whether an individual will be in a group that reduces more quickly over time—or the inverse, factors that affect low reduction success with the EC—could assist in the development of harm-reduction strategies that would benefit both CC and EC users who would like to quit. Additional research, especially on larger samples, is critical to understanding the reduction and switching patterns amongst smokers using ECs as a stepping-stone on the way to cessation. 

## Figures and Tables

**Figure 1 ijerph-19-07452-f001:**
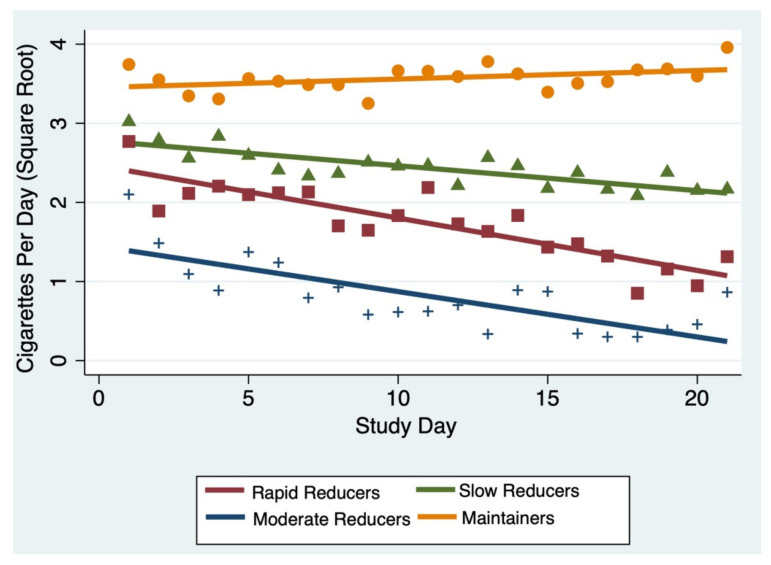
Graph of the four separate CPD reduction trajectories, identified using Nagin clustering. The number and order of classes were determined using Nagin clustering. Number and order of classes tested ranged from 1–4. Shapes correspond with the color of group assignment; for example, orange circles are those individuals assigned to the “Maintainers” group.

**Figure 2 ijerph-19-07452-f002:**
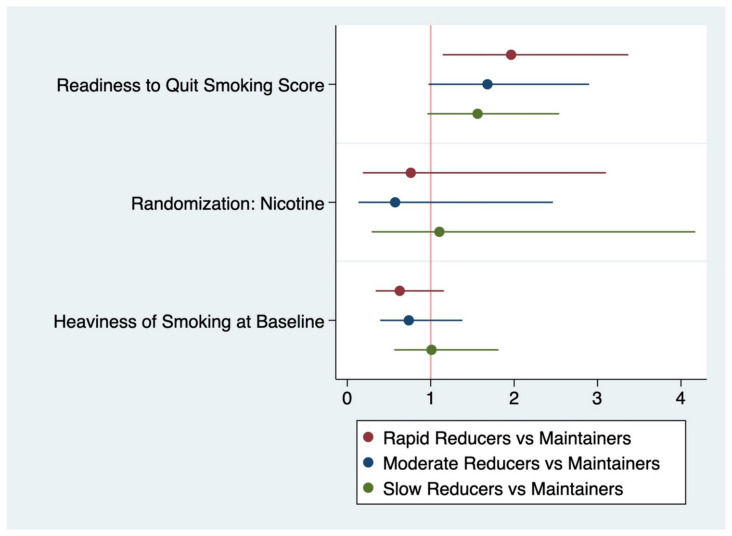
Representation of relative risk ratios with confidence intervals from the composite multinomial logistic regression model predicting trajectory class assignment. All covariates in final model included here. The whiskers represent 95% confidence intervals for each coefficient estimate.

**Table 1 ijerph-19-07452-t001:** Equation of lines for each trajectory group identified from Nagin clustering. The number and order of classes tested ranged from 1–4; optimal model fit was determined to be four linear classes.

Class (*N*, %)	CPD at Start ^1^	Linear Coefficient	Standard Error
Rapid Reducers (23, 27%)	6.3	−0.07	0.01
Moderate Reducers (18, 21%)	2	−0.06	0.009
Slow Reducers (27, 32%)	7.8	−0.03	0.009
Maintainers (16, 19%)	12.3	0.01	0.009

^1^ While all participants reported smoking at least 10 CPD on average prior to enrolling in the study, some participants already began reducing their cigarette intake prior to the first day of data collection. This value is representative of the cigarettes smoked using EMA-gathered smoking data beginning on study day 1.

**Table 2 ijerph-19-07452-t002:** Sample characteristics, stratified by cigarettes per day reduction trajectory group assignment.

	Total	Outcome = Trajectory Group			
	*N* (%) or Mean (SD)	Rapid Reducers *N* (%) or Mean (SD)	Moderate Reducers *N* (%, SD)	Slow Reducers N (%, SD)	Maintainers *N* (%, SD)
*Demographics*					
Age in years	28.4 (3.9)	28.5 (3.8)	28.3 (4.2)	28.3 (4.0)	28.4 (4.0)
Gender					
Male	57 (67.9)	13 (56.5)	16 (88.9)	20 (74.1)	8 (50.0)
Female	27 (32.1)	10 (43.5)	2 (11.1)	7 (25.9)	8 (50.0)
Education					
High school or less	20 (23.8)	6 (26.0)	4 (22.2)	7 (25.9)	3 (18.8)
Some college	33 (39.3)	10 (43.5)	10 (55.5)	7 (25.9)	6 (37.5)
College or post-graduate	31 (36.9)	7 (30.4)	4 (22.2)	13 (48.1)	7 (38.9)
Race/Ethnicity					
Non-Hispanic African American/Black	20 (24.1)	5 (21.7)	6 (33.3)	4 (14.8)	5 (31.2)
Non-Hispanic white	28 (33.7)	10 (43.5)	3 (16.7)	9 (33.3)	6 (37.5)
Other non-Hispanic	14 (16.9)	2 (8.7)	3 (16.7)	8 (29.6)	1 (6.3)
Hispanic of any race	21 (25.3)	5 (21.7)	6 (33.3)	6 (22.2)	4 (25.0)
*Treatment Assignment*					
Control (Placebo)	42 (50.0)	11 (47.8)	10 (55.6)	13 (48.1)	8 (50.0)
Active Nicotine	42 (50.0)	12 (52.2)	8 (44.4)	14 (51.9)	8 (50.0)
*Tobacco Use*					
Baseline CPD					
10	22 (26.2)	7 (30.4)	6 (33.3)	8 (29.6)	1 (6.3)
>10	62 (73.8)	16 (69.6)	12 (66.7)	19 (70.4)	15 (93.8)
Heaviness of Smoking Index (0–5 scale) *	2.6 (1.2, 0–5)	2.2 (1.1)	2.5 (0.9)	2.8 (1.2)	2.9 (1.4)
Made serious quit attempts (>1 day) in last year	41 (48.8)	14 (60.9)	9 (50.0)	12 (44.4)	6 (37.8)
Readiness to Quit (1–10 scale, 1–8 apply to current smokers)	5.56 (1.4)	6 (1.1)	5.8 (1.4)	5.7 (1.6)	4.8 (1.1)
Confidence in Quit Ability	6.4 (2.6)	6.7 (2.6)	7.2 (2.6)	6.3 (2.5)	5.0 (3.2)
Smoking behavioral dependence scale (11 items)					
Mild	15 (17.9)	2 (8.7)	5 (27.8)	7 (25.9)	1 (6.3)
Moderate	42 (50.0)	11 (47.8)	10 (55.5)	14 (51.9)	7 (43.8)
Strong to very strong	27 (32.1)	10 (43.5)	3 (16.7)	6 (22.2)	8 (50.0)
*Health Behavior Outcomes*					
Week 1 ECPD	1.06 (0.74)	1.0 (0.64)	1.2 (0.61)	1.01 (0.96)	0.94 (0.65)
Week 3 ECPD	0.99 (1.5)	0.99 (1.5)	0.92 (0.70)	1.1 (0.66)	0.68 (0.47)
50% Smoking reduction obtained at week 3 **	47 (59.5)	14 (60.9)	16 (88.9)	15 (55.6)	2 (12.5)

* *p* < 0.05, ** *p* < 0.01. *p*-Values obtained using binomial multinomial logistic regression.

## Data Availability

The data presented in this study are available on request from the corresponding author.

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
