# Peer review of "Analyzing Trajectories of Acute Cigarette Reduction Post-Introduction of an E-Cigarette Using Ecological Momentary Assessment Data"

_ijerph, 2022, doi:10.3390/ijerph19127452_

Round 1
Reviewer 1 Report
This MS describes a secondary analysis of data from a trial of active vs. placebo ecigs to reduce cigarette smoking over 3 weeks. The authors then classified participants into multiple trajectories based on behavior over the 3 weeks. They report that predictors of being in more successful classes in terms of cig reduction included readiness to quit, previous quitting, and lower CPD at baseline. They also conclude that ecigs may promote cig reduction. The potential importance of the study is that it addresses an important issue, namely whether vaping can help cigarette smokers to quit. It also provides more detailed data than many previous studies of the issue. Enthusiasm for the paper is dimmed somewhat by some concerns about the literature synthesis and particularly by the question of whether the data may be obsolete because the ecig technology is no longer widely used. This issue should be more clearly addressed. Most importantly, the MS suggests multiple times that the data are consistent with the potential utility of EC to reduce CC use, or words to that effect, but it's not clear that conclusion matches the findings.
Comments
- The sentence starting on line 42 is a bit strong in suggesting that research has demonstrated that ecigs can help. I think suggested is a more appropriate term. Additionally, the 6 references for that sentence are all based on older ecig technology that is mostly obsolete. Similarly, at the end of the paragraph the authors refer to mechanisms that explain transition from cigs to ecigs but I think work remains to establish that this is actually what occurs. In general the authors would be well-served to think of the science as less settled on some of these points.
- No hypotheses are reported.
- Suggest reporting when data were collected.
- I appreciate that the authors have already published some of the information on the sample elsewhere but it would be helpful to better characterize the sample. How heavy was their smoking at BL? Table 2 doesn't make this entirely clear as they are listed in 2 categories - does this mean all smoked at least 10 cpd? It also seems worthwhile to repeat here any inclusion/exclusion criteria related to vaping, and if relevant whether the sample were using ecigs at baseline, or had in the past.
- The cig and ecig variables used are somewhat unclear. It appears that the cig predictor was self-reported cigs, but it also appears that this was measured 4x each day, and it is unclear which measure was used. Similarly, it's uncertain whether e-cig frequency was based on self-report or on used devices. In both cases, it's unclear whether these measures were repeated daily or at a different frequency.
- Intent-to-treat refers to including all randomized participants but does not require imputing values that are missing, as is reported here. If attrition differed as a function of the predictors of interest, then imputing values may increase bias unless the likelihood of attrition is highly correlated with the imputed outcome. It is not clear whether the authors tested the potential impact of this imputation. In most cases, analysis of available data is actually superior to imputation - see eg https://www.sciencedirect.com/science/article/pii/S2451865417301941. Additionally it is unclear what "recoding all reduction outcomes as zeroes" means - imputing no change for those participants?
- Minor comment, in some passages the paper is inconsistent in referring to participants in singular or plural forms - eg, lines 80-82.
- In table 2, the proportions seem to be done in the opposite way from what I would expect - the % shown parenthetically are the % of the subgroup within each trajectory, rather than vice versa, if I am understanding correctly. That is, for example, there were 13 men in the rapid reducer group, which is listed as 22.8%; that seems to mean that this was 22.8% of the men. I think it would be more intuitive to list the % of rapid reducers who were men , and the same for the rest of the rows.
- Whether or not participants were paid for their time or contingent on cessation should be reported (I now see this is mentioned in passing in the discussion - should be in the methods).
- On lines 279-283, the rationale for the statement that these findings "are consistent with literature that ECs can help with CC cessation reduction" is unclear to me. It does not appear that any modeled variables related to EC use had a meaningful impact on CC outcomes. Similarly, on lines 336-338, the authors make a statement suggesting that EC helped some participants reduce CC intake but this does not seem consistent with the results.
- Most importantly, the MS is focused on ecigs, and whether they may be a useful device for helping cig smokers to reduce cig intake. However, as far as I can tell there is no indication that ecig use or ecig group assignment had any impact. The predictors of cig use and trajectory membership are entirely constructs that we would expect in a traditional cessation trial - baseline dependence and related measures, readiness to quit.
Reviewer 2 Report
Cigarettes smoke is a global problem associated with many seriuos diseases. Electronic cigarettes are one of the method uses as smoking cessation treatment.
Unfortunately, until now it’s little to know about usefullness of electronic cigarettes to reduce or quit smoking, therefore in my opinion this paper is valuable.
I do however, some concerns.
- In my opinion the conclusions of the study is too heavy: that „clinicians can more effectively advise their patients on EC-based smoking cessation and determine whether a given patient may be a “good fit” for ECs as a smoking cessation device” (verse 28-30). Firstly, the results are based on participants self-reported, which potentially can be associated with errors. Secondly, the groups were small, therefore the study should be treated as preliminary.
Third, participants who reported smoking 10 or > 10 CPD were enrolling in the study. How many cigarettes were smoked in all subgroups?
- It is important, beacuse participants who smoke 2 packages are probably more addicted than who smoke 10-15 cigarettes per day.
- Why authors chose 4.5% nicotine in ECs? The average concentration of nicotine contains in cigarette is 10-12 mg and the authors used 4.5% (4.5 g/100 mL)? Please introduce some details about ECs. What kind of cigarettes were smoked by the participatns (mean concentration of nicotine)? What was the average concentration of nicotine in one combustible cigarette?
- In my opinion the eligibility criteria should be described. Authors claimed that it was published in paper number 14 but this paper is from 2016 year so it will be well to recall those data.
- Also readness to quit between groups was not statistically significant (p=0.05). The authors correct mentioned that statistically significant differences is when p value is lower than 0.05 (verse 149) (or even <0.01 in clinical study), not equal 0.05.
- One drop of rapid reducer is linked to the linear curve of slow reducer. If it is not an error, please extend the distances between the curves.
- In the subgroups of different type reducers were included both participants who took placebo and ECs?
Author Response
See word document attached. Thank you!

Round 2
Reviewer 1 Report
The authors have been relatively responsive to the initial review. Some concerns remain:
1. The response letter states that hypotheses have been added, but these aren't truly hypotheses. What trajectory groups were predicted? What common characteristics were expected to be lumped together?
2. In their response the authors write that they "agree with the premise" that imputing missing data is typically an inferior approach allowing it to remain missing, but they continue to impute, and to incorrectly describe this as an intent-to-treat approach. I think they should either not impute, or make some effort to evaluate the potential impact of imputation. Because they have data to enable this, simply adding that the additional optional procedure may introduce bias to the limitations section seems insufficient.
3. It remains unclear what this manuscript would contribute to the literature. The original review indicated that the predictors of trajectory membership appear unrelated to e-cig use. Predictors of cigarette reduction included baseline heaviness, readiness to quit - these are well-known predictors of quitting success. The authors have not satisfactorily addressed this concern.
Author Response
Please see attached word document for response to Reviewer 1. Thank you.

Reviewer 2 Report
In currect version I accept the paper
Author Response
NA - we thank the reviewer for his helpful comments in the first round of review.